

# Two new species of *Eleutherodactylus* from western and central Mexico (*Eleutherodactylus jamesdixoni* sp. nov., *Eleutherodactylus humboldti* sp. nov.)

Thomas J. Devitt, Karen Tseng, Marlena Taylor-Adair, Sannidhi Koganti, Alice Timugura and David C. Cannatella

Freshman Research Initiative and Department of Integrative Biology, University of Texas at Austin, Austin, Texas, United States

## ABSTRACT

**Background:** The subgenus *Syrrhophus* (genus *Eleutherodactylus*) contains >40 species of small, direct-developing frogs that occur at low to moderate elevations from Texas through Mexico and into Guatemala and Belize, with two species in western Cuba. Morphological conservatism and phenotypic convergence have made species delimitation challenging and resulted in a complicated taxonomic history. Since 2015, molecular systematic work has uncovered eleven new species from western Mexico and one from eastern Mexico, but current taxonomy still underestimates species level diversity and there is confusion surrounding the validity and boundary of several species.
**Methods:** We used phylogenetic analysis of 16S ribosomal RNA (rRNA) sequences, multivariate statistical analysis of morphological data, and bioacoustic analysis of male advertisement calls to discover two additional unnamed species of *Eleutherodactylus* from Central and Western Mexico. We describe those species here.
**Results:** *Eleutherodactylus* (*Syrrhophus*) *humboldti* sp. nov. is described from the Quaternary Valle de Bravo volcanic field of the Eje Neovolcánico in Central Mexico. This species is sister to *E. maurus* and is 3% divergent in 16S. *Eleutherodactylus* (*Syrrhophus*) *jamesdixoni* sp. nov. is described from the Sierra Madre Occidental of western Mexico. This species is sister to *E. nitidus* and is 3% divergent. We provide color photographs, advertisement call recordings, and molecular diagnoses of these new species and their sister species to aid future workers.

## INTRODUCTION

About half of the world's >8,500 amphibian species occur in the New World tropics and subtropics (AmphibiaWeb, https://amphibiaweb.org/index.html). A third of those comprise a diverse radiation (>1,200 named species) of frogs that breed on land and have direct development, forgoing the tadpole stage. Ranging from the southern United States

Corresponding author
Thomas J. Devitt, tdevitt@utexas.edu

[1] Hedges, Duellman, and Heinicke (2008:21) coined Terrarana as an unranked taxon name above the family-group level. *Dubois (2009)* argued that the name should be emended to Terraranae, because "class-series" taxonomic names should be formed in the nominative plural. However, (i) the term class-series was coined by Dubois, and is not generally recognized by taxonomists, and (ii) the International Code of Zoological Nomenclature does not govern taxon names above the family level (as *Dubois (2009)* admits). Thus, it is unnecessary to emend the name, and doing so causes confusion. In the interest of stability, we retain the original spelling.

to northern Argentina and throughout the West Indies, members of the clade Terrarana[1] (*Hedges, Duellman & Heinicke, 2008*) are some of the most ubiquitous and abundant of all Neotropical amphibians, yet also among the least well known. Most species of Terrarana were formerly considered members of a single, large genus (*Eleutherodactylus*), but recent taxonomic revisions based primarily on phylogenetic analysis of DNA sequence data have provided a better understanding of evolutionary history and a predictive classification for this clade (*Crawford & Smith, 2005*; *Heinicke, Duellman & Hedges, 2007*; *Hedges, Duellman & Heinicke, 2008*; *Heinicke et al., 2018*; *Barrientos et al., 2021*).

Within Terrarana, the Mesoamerican subgenus *Syrrhophus* (genus *Eleutherodactylus*) (*Hedges, Duellman & Heinicke, 2008*) has received less attention than other clades until about the last 8 years. Ranging continuously from west-central Texas throughout Mexico and into Guatemala and Belize, with two species in western Cuba, *Syrrhophus* species occupy a wide variety of different biomes along elevational gradients from sea level to 2,400 m including desert, savanna, tropical and subtropical forest, and montane woodland. Although distributional limits for most species are not well known, several species are thought to occur only at the type locality or its vicinity. At least three species may be found in sympatry in several areas. Hybridization has never been reported, although some subspecies have been recognized based on presumed zones of intergradation (*Dixon, 1957*; *Duellman, 1958*; *Lynch, 1970*).

Species in this group were formerly assigned to the genera *Syrrhophus* and *Tomodactylus*, generally corresponding to eastern and western species, respectively, and the two genera were thought to be distinguishable based on the presence (in *Tomodactylus*) or absence (in *Syrrhophus*) of a lumbar gland (*Smith & Taylor, 1948*). *Lynch (1968*, *1970*, *1971)* continued to recognize these genera as distinct, and discussed morphological differences thought to distinguish them from *Eleutherodactylus*. *Hedges (1989)* proposed that species of *Syrrhophus* and *Tomodactylus* be placed in the subgenus *Syrrhophus* based on osteological characters. *Heinicke, Duellman & Hedges (2007)* recovered two Cuban species (*Eleutherodactylus* (*Euhyas*) *symingtoni* and *E. zeus*) with mainland species in their phylogenetic analysis of mitochondrial and nuclear DNA sequences, suggesting mainland *Syrrhophus* originated by oceanic dispersal from Cuba during the Miocene ≈19 Mya (*Hedges, 1989*; *Haas & Hedges, 1991*; *Heinicke, Duellman & Hedges, 2007*) but see (*Duellman, 1970*). The Antillean ancestor may have dispersed to Mexico *via* the Yucatan Peninsula, or from the Caribbean into Eastern North America as some fossil evidence suggests (*Holman, 1967*; *Crawford & Smith, 2005*).

*Hedges, Duellman & Heinicke (2008)* conducted a monographic revision of Terrarana using DNA sequences, including three mainland and the two Cuban species of *Syrrhophus* in their phylogeny. They recognized the mainland and Cuban clades as "species series", the *E. longipes* series (24 species) and *E. symingtoni* series (two species), respectively. Within the *E. longipes* series, species were placed into six species groups (*E. leprus*, *E. longipes*, *E. marnockii*, *E. modestus*, *E. nitidus*, and *E. pipilans*) first defined by *Lynch (1970)* and later revised by *Hedges (1989)* and *Hedges, Duellman & Heinicke (2008)*.

In the last eight years, workers have described 11 new species from western Mexico (*Reyes-Velasco et al., 2015*; *Grünwald et al., 2018*, *2021*; *Palacios-Aguilar & Santos-Bibiano,*

*2020*) and one from eastern Mexico (*Hernández-Austria et al., 2022*). However, confusion remains surrounding the validity and boundaries of some species and their phylogenetic relationships. Here, we describe two new species from Central and Western Mexico using mitochondrial DNA sequences, morphology, and acoustic analysis of male advertisement calls from topotypic samples. Where we have not sampled topotypes or are uncertain about species identification, we indicate that uncertainty to prevent future workers from propagating erroneous identifications and further confusing the taxonomy of this group.

## MATERIALS AND METHODS

### Ethical statement

Tissue samples were donated from natural history museum collections in Mexico that were made with permission from the Secretaría de Medio Ambiente y Recursos Naturales (permit numbers 02729, FAUT-0110, and FAUT-0243). Collections in 2022 followed Animal Use Protocol AUP-2022-00192, issued by the Institutional Animal Care and Use Committee at the University of Texas at Austin.

### mtDNA sequence data collection and phylogenetic analysis

DNA was isolated using spin columns (Qiagen, Hilden, Germany) and quantified using a Qubit v1 fluorometer with a broad range dsDNA assay (Thermo Fisher, Waltham, MA, USA). We amplified and sequenced an ~570 base-pair (bp) fragment of the 16S ribosomal RNA gene in the forward and reverse directions using the primers 16Sar and 16Sbr (*Palumbi, 1996*). Specimens and associated metadata are provided in Table S1. We used Geneious Prime 2020.2.3 (Biomatters Ltd., Auckland, New Zealand) to create assemblies and consensus sequences; bases with >1% chance of an error were automatically trimmed.

To estimate phylogenetic relationships, we assembled a data matrix of 360 16S sequences of 595 characters of which 183 sequences were generated by us and 177 downloaded from GenBank (Table S1). *Eleutherodactylus* (*Syrrhophus*) *zeus* and *E.* (*Syrrhophus*) *symingtoni* were designated as outgroups. Sequences were aligned with MAFFT version 7.450 using the *einsi* alignment strategy (*Katoh et al., 2002*; *Katoh & Standley, 2013*) and then manually adjusted in Mesquite 3.2 (*Maddison & Maddison*) (Data S1).

We used PAUP* (*Swofford, 2003*) to extract the shared derived character-states for each species in the 360-tip matrix. Character-state changes that were unambiguous under both ACCTRAN and DELTRAN algorithms were extracted by optimizing characters on the best likelihood tree using the command "*describe 1/plot = clado labelnode = y diagnose = n apolist = y*".

We pruned the 360-tip matrix to include only unique sequences, yielding 160 sequences for the likelihood analysis (Fig. 1); sequences from all species were retained (Data S1). IQ-TREE v.2.1.2 (*Minh et al., 2020*) was used to select the optimal model of evolution (*-m TEST*) under the Bayesian Information Criterion (*Kalyaanamoorthy et al., 2017*) and to estimate the phylogeny using maximum likelihood. Base frequency parameters (*-mfreq F, FO*) and rate heterogeneity parameters (*-mrate E, G*) were estimated. IQ-TREE2 was run ten times to check for consistency of the log-likelihood; all runs yielded the same topology

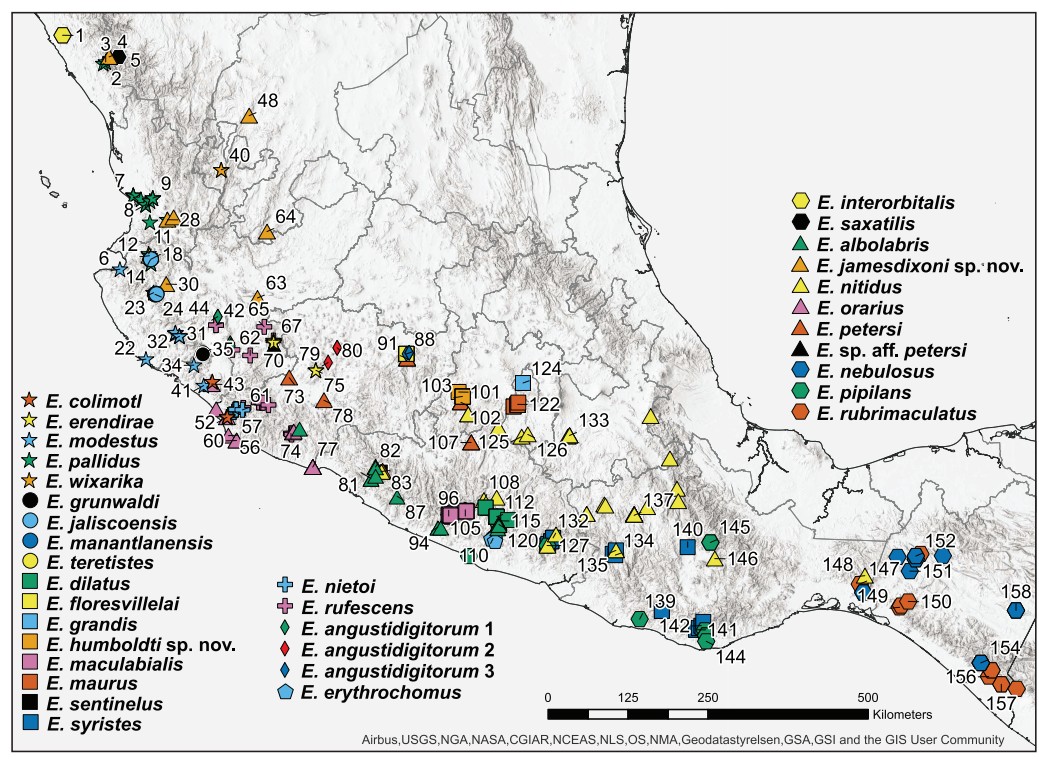

**Figure 1** Map of 160 samples used in phylogenetic analysis.

and similar log-likelihood values (within a range of one likelihood unit). Ten thousand ultrafast bootstrap replicates (*Minh, Nguyen & von Haeseler, 2013*; *Hoang et al., 2017*) were used to assess branch support. Bootstrap proportions were mapped onto the optimal likelihood tree which was visualized and annotated using the Interactive Tree of Life (iTOL) online tool (*Letunic & Bork, 2019*).

We calculated mean pairwise p-distances (uncorrected) between species with the *ape 5.6-2* package (*Paradis & Schliep, 2018*) and a custom R script written in RMarkdown (Script S1) using the pairwise deletion for missing data option.

## Specimens examined and morphometric analysis

We examined type specimens or photos of type specimens of *Eleutherodactylus albolabris* (*Taylor, 1943*), *E. maurus* (*Hedges, 1989*), *E. modestus* (*Taylor, 1942*), *E. nebulosus* (*Taylor, 1943*), *E. nitidus* (*Peters, 1870*), *E. orarius* (*Dixon, 1957*), *E. petersi* (*Duellman, 1954*), *E. pipilans* (*Taylor, 1940a*), *E. rubrimaculatus* (*Taylor & Smith, 1945*), *E. rufescens* (*Duellman & Dixon, 1959*), *E. syristes* (*Hoyt, 1965*) and *E. teretistes* (*Duellman, 1958*). For more recently described western species we compared specimens to diagnoses (*Reyes-Velasco et al., 2015*; *Grünwald et al., 2018*, *2021*; *Palacios-Aguilar & Santos-Bibiano, 2020*).

We photographed preserved adult specimens using a DSLR camera mounted overhead on a copy stand and with a 0.5 mm precision ruler for scale. The ruler was elevated into the same focal plane as the part of the specimen being photographed. Images were imported

into ImageJ2 (*Rueden et al., 2017*) and the scale was set by using the straight-line selection tool to measure 10 mm on the ruler in the image. We used the same tool to measure 17 variables on preserved specimens after *Watters et al. (2016)*, including head width (HW), snout-vent length (SVL), tibia length (TL), interorbital distance (IOD), eye diameter (ED), internarial distance (IND), eye-nostril distance (EN), foot length (FL), tympanum diameter (TD), thigh length (THL), snout length (SL), hand length (HAL), forearm length (FLL), finger III length (Fin3L), finger III disk width (Fin3DW), finger IV length (Fin4L), and finger IV disk width (Fin4DW). One of us (SK) collected all measurements to avoid intermeasurer bias (*Lee, 1982*). Images are provided in the Dryad digital repository associated with this article (*Devitt et al., 2023*).

We explored morphological variation using principal components analysis (PCA) on two separate datasets. The *E. maurus* dataset (Data S2) consisted of seven topotypic *E. maurus* and eight individuals of its sister taxon (which we describe below as *E. humboldti* sp. nov.) identified in our phylogenetic analysis. The *E. nitidus* dataset (Data S3) consisted of five topotypic *E. nitidus* and three individuals of its sister species (*E. jamesdixoni* sp. nov.). We first tested whether variables were skewed using the *dlookr* package for R (*Ryu, 2022*). Eleven variables in Data S3 (SVL, IND, FL, TD, THL, SL, HAL, FLL, Fin3L, Fin3DW, and Fin4DW) and seven variables in Data S3 (SVL, IOD, IND, EN, THL, SL, and Fin3DW) were skewed so we transformed all measurements on a log base 2 scale. We performed PCA separately for each dataset on the correlation matrix using the R packages *tidyverse* (*Wickham et al., 2019*) and *broom* (*Robinson, 2014*) (Script S2).

## Call analysis

We recorded advertisement calls from topotypic samples of *E. maurus* ($n$ = 4 individuals), *E. nitidus* ($n$ = 4), *E. humboldti* ($n$ = 3) and *E. jamesdixoni* ($n$ = 3). Advertisement calls were recorded in waveform audio file format (.wav) using a Marantz PMD661 handheld recorder with a Sennheiser shotgun microphone at a sampling rate of 44.1 kHz, held 25–50 cm from the frog. We first filtered background noise from recordings using the noise reduction feature in Audacity 3.2.1 (*Audacity Team, 2022*). Filtered recordings were then analyzed using Raven Pro 1.6.3 (*Cornell Lab of Ornithology, 2022*). We selected portions of recordings for analysis where the rate of the calls had stabilized; if a recording contained an intercall interval longer than three times that of the average, we deemed that interval to be a break between calling bouts. Calls were classified as pulsatile (poorly defined energy bursts without intermittent silence) or pulsed (a call consisting of well-defined energy bursts) following *Köhler et al. (2017)*. For all calls, we measured the peak amplitude, call duration, intercall interval duration, call rise time, dominant frequency, and maximum and minimum fundamental frequency. If calls were pulsed, we also measured pulse duration and interpulse interval. Definitions for these characteristics follow *Köhler et al. (2017)*, apart from call rise time which follows *Cocroft & Ryan (1995)*. Descriptive statistics for call characteristics were calculated in R 4.2.2 (*R Core Team, 2022*). We used Seewave 2.2.0 (*Sueur, Aubin & Simonis, 2008*) implemented in R to plot the temporal and spectral properties of the signals using short-time Fourier transform (STFT) calculations. To avoid subjectivity in call descriptions and to facilitate comparison with other species, we provide
recordings (raw and filtered) and associated output files in the Dryad digital repository for this article (*Devitt et al., 2023*).

## Species delimitation

We view species as separately evolving lineages (*Simpson, 1951*; *Wiley, 1978*; *de Queiroz, 1998*) and the evolutionary processes underlying lineage divergence (natural selection, mutation, gene flow, and genetic drift) as contingent properties that may be used as evidence of independence. For many species, evidence of barriers to gene flow based on genetic data are useful in assessing whether they are evolving separately. We agree with *Renner (2016)* and *Vences (2020)* in emphasizing diagnosis over description to accelerate the cataloging of species level diversity, and use DNA-based diagnoses, color images, advertisement call recordings, and geographic distribution to diagnose these species, and include a list of DNA synapomorphies for other named western *Syrrhophus* species.

## New zoological names

The electronic version of this article in Portable Document Format (PDF) will represent a published work according to the International Commission on Zoological Nomenclature (ICZN), and hence the new names contained in the electronic version are effectively published under that Code from the electronic edition alone. This published work and the nomenclatural acts it contains have been registered in ZooBank, the online registration system for the ICZN. The ZooBank LSIDs (Life Science Identifiers) can be resolved, and the associated information viewed through any standard web browser by appending the LSID to the prefix http://zoobank.org/. The LSID for this publication is urn:lsid:zoobank. org:pub:C59B3FB8-839F-407C-ADC0-67C1FEA6A5BB. The online version of this work is archived and available from the following digital repositories: PeerJ, PubMed Central SCIE and CLOCKSS.

## RESULTS

### Alignment

The alignment of 160 unique sequences had 595 characters of which 170 were parsimony-informative and 382 were constant. The best model of molecular evolution was HKY+FO+G4 (three classes of substitutions, AC=AT, AG=CT, CG=GT; unequal base frequencies; base frequencies optimized under likelihood; and discrete gamma model with four rate categories).

### Phylogeny

The log-likelihoods of the ten best trees ranged from −4,996.8648 to −4,997.5458. The best tree is shown in Fig. 2 and Fig. S1. Parameter estimates for the best tree are: Rate parameters AC=AT=CG=GT 1.0000, AG=CT 9.2120; base frequencies A: 0.375, C: 0.252, G: 0.114, and T: 0.259; alpha parameter of gamma distribution 0.195.

For almost all species, the tips form monophyletic groups with ultrafast bootstrap support >90; however, the support value for *E. petersi* is 82. Two clusters of sequences are not monophyletic. The cluster of *E. pipilans* sequences is paraphyletic to a clade (bootstrap

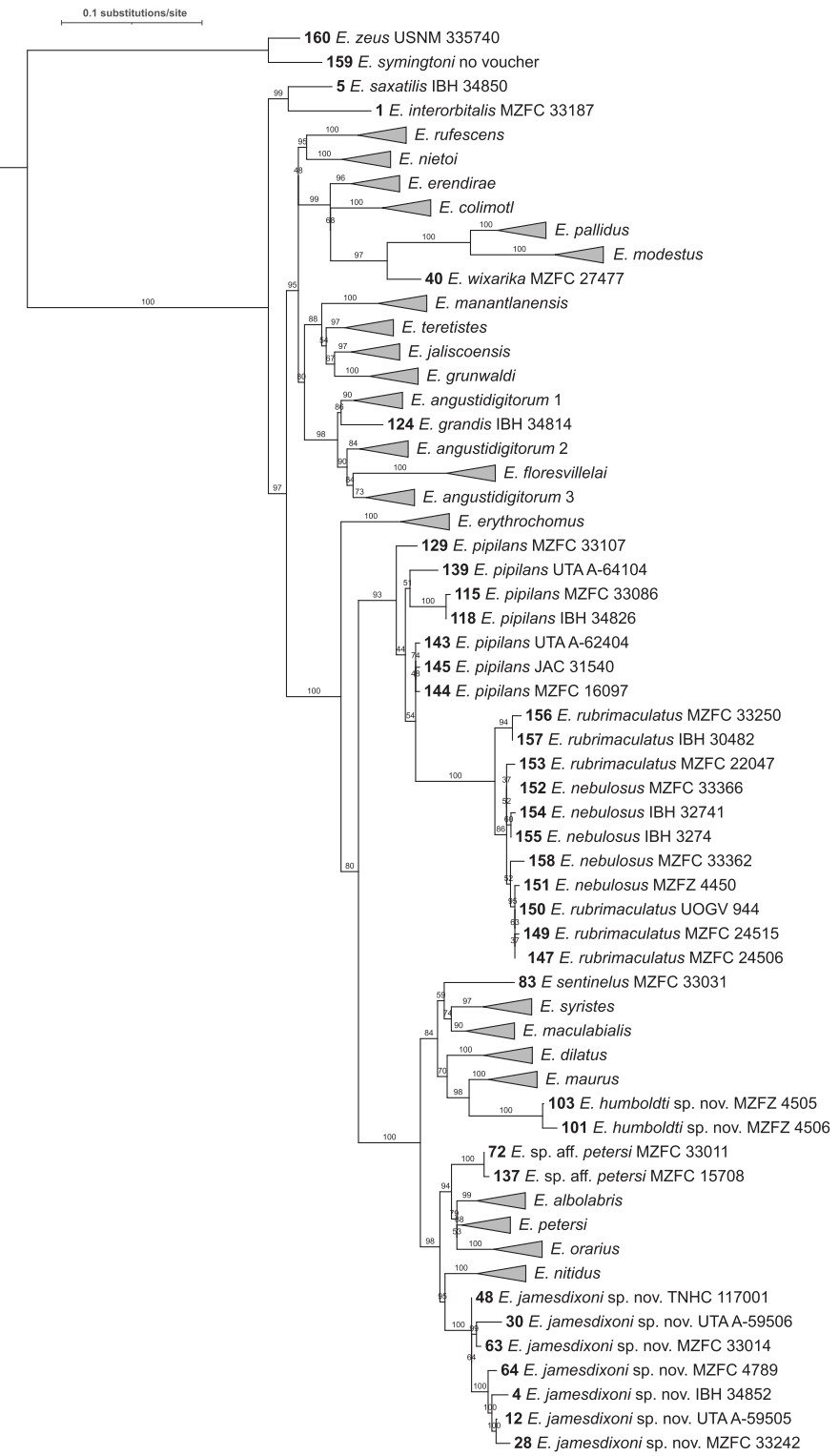

**Figure 2 Maximum likelihood tree of 160 sequence dataset from 10 replicated runs.** Numbers on nodes represent ultrafast bootstrap support values from 10,000 ultrafast bootstrap replicates. Monophyletic groups of sequences other than the two new species described here have been collapsed to a single tip for clarity; the full tree is shown in Fig. S1. Acronyms are as follows: IBH, Colección Nacional de Anfibios y Reptiles, Instituto de Biología, Universidad Nacional Autónoma de México; JAC, Jonathan A.

**Figure 2** (continued)
Campbell, uncatalogued specimen; MZFC, Museo de Zoología, Facultad de Ciencias, Universidad Nacional Autónoma de México; MZFZ, Facultad de Estudios Superiores Zaragoza, Universidad Nacional Autónoma de México; TNHC, Texas Natural History Collections, the University of Texas at Austin; USNM, United States National Museum of Natural History; UTA A-, Amphibian Collection, the University of Texas at Arlington; and UOGV, Uri Omar García Vázquez, uncatalogued specimen. See Table S1 for full metadata.              

support 100) made up of *E. rubrimaculatus* and *E. nebulosus* sequences. The clade including *E. pipilans*, *E. rubrimaculatus*, and *E. nebulosus* has high bootstrap support (94). However, support for the node that makes *E. pipilans* paraphyletic is 86.

A clade containing *Eleutherodactylus grandis* (*Dixon, 1957*), *E. angustidigitorum* (*Taylor, 1940b*) (named with suffixes 1, 2, and 3), and *E. floresvillelai* (*Grünwald et al., 2018*) has a bootstrap support value of 93, but *E. angustidigitorum* is paraphyletic with respect to *E. grandis* and *E. floresvillelai*. Support for the nodes that make *E. angustidigitorum* paraphyletic is 92 (*E. angustidigitorum* 1 + *E. grandis*) and 74 (*E. angustidigitorum* 3 + *E. floresvillelai*).

The level of support for groups of species varies. Support for the six deepest nodes ranges from 91–100. Support for sister-species ranges from 42 (*E. dilatus* and *E. syristes*) to 100 (*E. modestus* and *E. pallidus*). Support for each of the two new species described below and their sister-species are 96 (*E. maurus* and *E. humboldti* sp. nov.) and 78 (*E. nitidus* and *E. jamesdixoni* sp. nov.).

## Diagnostic characters

The number of unambiguous synapomorphies for a species ranged from 1 (*E. petersi*) to 16 (*E. floresvillelai*; Table S2). The number of unambiguous synapomorphies is 10 for *E. humboldti*, and 5 for *E. jamesdixoni*; Table S2. Bootstrap support for the clade of nine *E. petersi* sequences is 82. Bootstrap support values for all clades with >1 diagnostic characters were >90. Of the 195 state changes in Table S2, 155 were transitions and 40 were transversions.

## Pairwise distances

Mean distances between the two Cuban outgroup species (*E. zeus* and *E. symingtoni*) and ingroup species ranged from 0.104–0.159. The smallest mean between-species distance (uncorrected) was between *E. albolabris* and *E. petersi* (0.0203; Table S3). Excluding the *E. angustidigitorum* sequences, 12 interspecific distance values were <0.03. The two new species that we describe below are each divergent from their sister taxon by mean uncorrected pairwise *p*-distances of 0.0297 (*E. maurus—E. humboldti*) and 0.0333 (*E. nitidus—E. jamesdixoni*).

## Morphometric analysis

For the *E. maurus* dataset, the first two principal components accounted for 65% of the total variance with the first PC explaining about 46%. PC axis one had strong positive loadings for SVL, IND, EN, THL, HAL, FLL, Fin3L Fin3DW, Fin4L, and Fin4DW, while PC axis two had strong positive loadings for HW, SVL, ED and IND, and strong negative
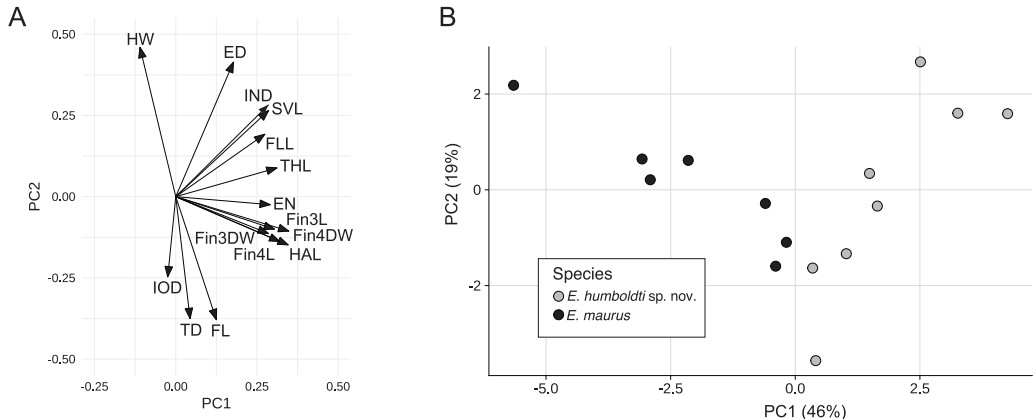

**Figure 3** Rotation matrix (A) and biplot of the fitted model (B) from principal components analysis of the *E. maurus* dataset (**Data S3**).

**Table 1 Factor loadings from the PCA of the *Eleutherodactylus maurus*—*E. humboldti* dataset (Data S3).** Values in bold indicate variables that have a strong (>0.24) effect on a given principal component. All variables were log base 2 transformed.

| Variable | PC1 | PC2 | PC3 |
|---|---|---|---|
| HW | **−0.110** | **0.460** | **0.004** |
| SVL | **0.286** | **0.265** | 0.078 |
| IOD | **−0.025** | **−0.247** | **−0.695** |
| ED | **0.178** | **0.414** | 0.038 |
| IND | **0.286** | **0.283** | **−0.352** |
| EN | **0.291** | −0.025 | **0.269** |
| FL | **0.125** | **−0.379** | −0.167 |
| TD | **0.044** | **−0.376** | **0.458** |
| THL | **0.312** | 0.089 | 0.147 |
| HAL | **0.346** | −0.148 | 0.005 |
| FLL | **0.274** | 0.192 | −0.115 |
| Fin3L | **0.348** | −0.106 | −0.071 |
| Fin3DW | **0.285** | −0.113 | 0.116 |
| Fin4L | **0.320** | −0.138 | −0.149 |
| Fin4DW | **0.304** | −0.100 | 0.003 |

loadings for FL and TD (Fig. 3A and Table 1). A biplot of the fitted model showed separation between *E. maurus* and *E. humboldti* along the first principal component axis (Fig. 3B).

For the *E. nitidus* dataset, the first two principal components accounted for 74% of the total variance, with the first PC explaining 57%. PC1 had strong negative loadings for HW, SVL, ED, EN, FL, THL, HAL, FLL, Fin3L, Fin4L and Fin4DW, while PC2 had strong negative loadings for IOD, IND, FLL, and Fin3DW (Fig. 4A and Table 2). A biplot of the

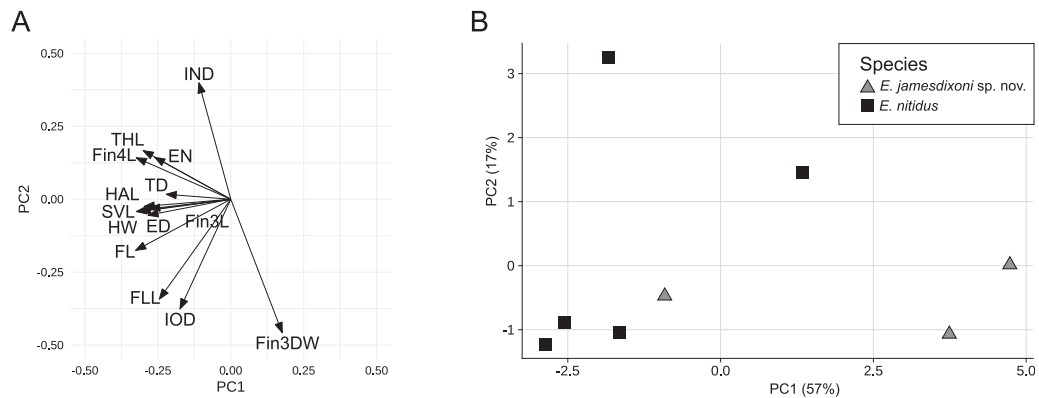

**Figure 4 Rotation matrix (A) and biplot of the fitted model (B) from principal components analysis of the *E. nitidus* dataset (Data S3).**

**Table 2 Factor loadings from the PCA of the *Eleutherodactylus nitidus—E. jamesdixoni* dataset (Data S3).** Values in bold indicate variables that have a strong (>0.24) effect on a given principal component. All variables were log base 2 transformed.

| Variable | PC1 | PC2 | PC3 |
|---|---|---|---|
| HW | **−0.284** | **−0.055** | **0.273** |
| SVL | **−0.324** | −0.042 | 0.136 |
| IOD | −0.174 | **−0.376** | **−0.291** |
| ED | **−0.278** | −0.033 | **0.244** |
| IND | −0.109 | **0.399** | 0.140 |
| EN | **−0.261** | 0.144 | **−0.330** |
| FL | **−0.326** | −0.175 | −0.015 |
| TD | −0.221 | 0.017 | **−0.533** |
| THL | **−0.300** | 0.167 | −0.177 |
| HAL | **−0.297** | −0.025 | 0.233 |
| FLL | **−0.245** | **−0.342** | **−0.254** |
| Fin3L | **−0.311** | −0.042 | 0.183 |
| Fin3DW | 0.176 | **−0.458** | −0.126 |
| Fin4L | **−0.324** | 0.142 | 0.076 |
| Fin4DW | **0.304** | −0.100 | 0.003 |

fitted model showed two of three *E. jamesdixoni* samples separating from *E. nitidus* samples along the first principal component axis (Fig. 4B).

## Call analysis

Temporal and spectral call properties are summarized in Table 3 for *Eleutherodactylus humboldti* and *E. maurus*. The calls of both species are pulsatile, but *E. humboldti* has a longer call, intercall interval, and call rise time as well as a lower dominant frequency compared to *E. maurus* (Fig. 5 and Table 3).

Peer J

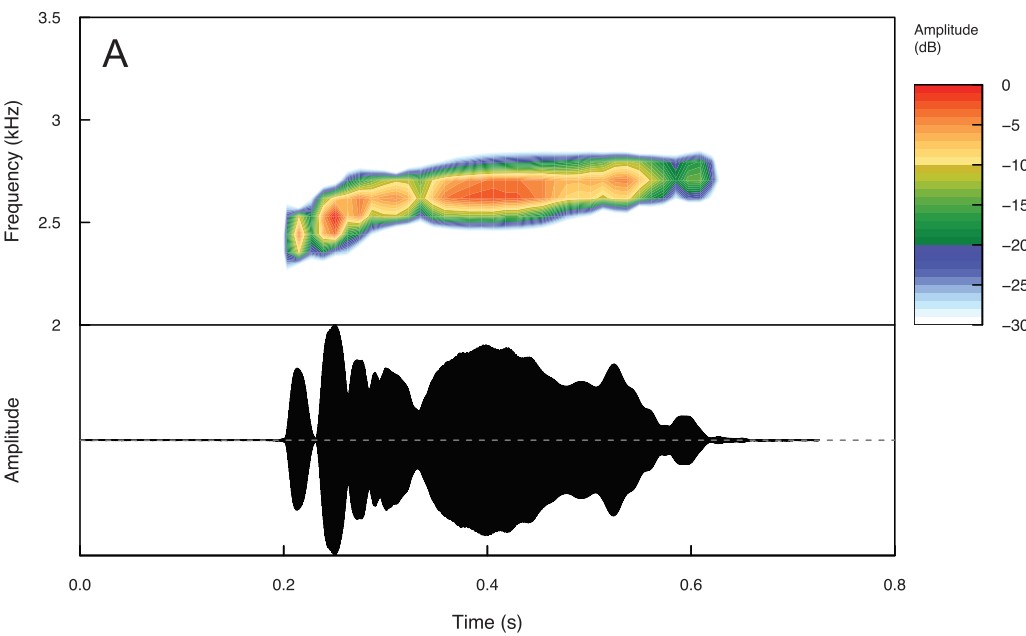

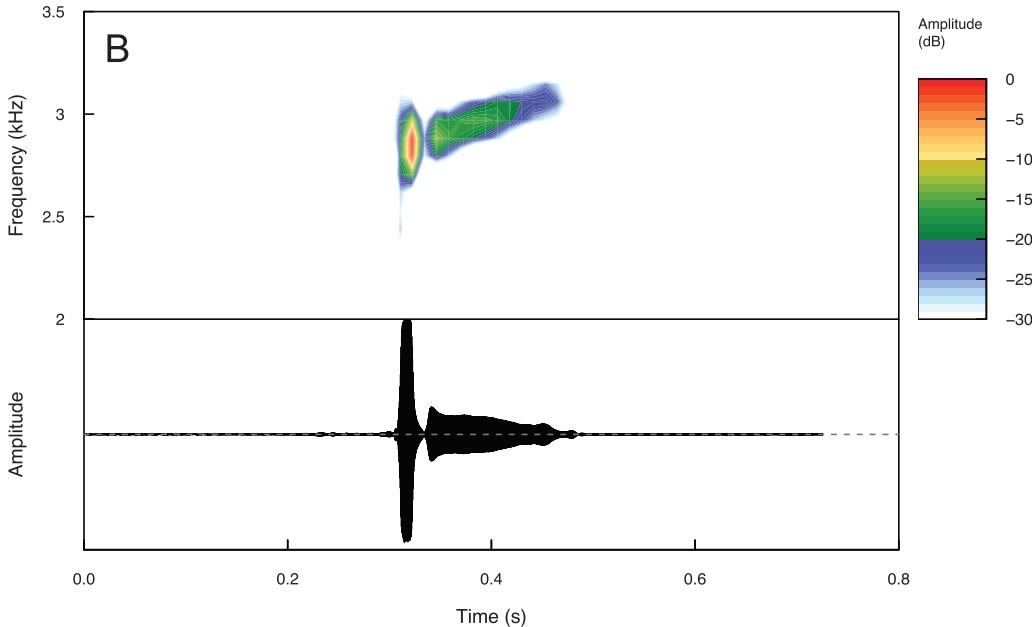

**Figure 5 Spectrograms and oscillograms of one call from (A) a paratype (MZFZ 4506) of *Eleutherodactylus humboldti* in comparison to (B) a topotypic *E. maurus* sample (IBH 34840) showing differences in temporal and spectral properties.**

Summary statistics for temporal and spectral call properties are summarized in Table 4 for *Eleutherodactylus jamesdixoni* and *E. nitidus*. Calls of *E. jamesdixoni* are pulsed and longer than calls of *E. nitidus*, which are pulsatile, but the dominant and fundamental frequencies are similar for both species (Fig. 6 and Table 4).

**Table 3 Comparison of call characteristics measured for *Eleutherodactylus humboldti* and *E. maurus*.** Means between species were compared using a Student's *t*-test.

| | E. humboldti sp. nov. (N = 3) | E. maurus (N = 4) | p value |
|---|---|---|---|
| Call duration (s) | 0.37 ± 0.05 (0.28–0.43; n = 23) | 0.16 ± 0.02 (0.13–0.20; n = 22) | p < 0.001 |
| Call rise time (s) | 0.14 ± 0.05 (0.04–0.29; n = 23) | 0.01 ± 0.00 (0.01–0.03; n = 22) | p = 0.004 |
| Peak amplitude | 5346 ± 2053 (2306–16780; n = 23) | 32716 ± 103 (31605–32768; n = 22) | p = 0.001 |
| Intercall interval (s) | 44.32 ± 1.44 (34.73–60.31; n = 20) | 28.57 ± 3.80 (21.00–37.05; n = 18) | p = 0.001 |
| Dominant frequency (kHz) | 2.57 ± 0.06 (2.50–2.67; n = 23) | 2.82 ± 76.27 (2.67–2.93; n = 22) | p = 0.006 |
| Minimum fundamental frequency (kHz) | 2.49 ± 75.39 (2.41–2.58; n = 23) | 2.71 ± 0.099 (2.58–2.84; n = 22) | p = 0.023 |
| Maximum fundamental frequency (kHz) | 2.64 ± 1.02 (2.50–2.76; n = 23) | 4.76 ± 2.53 (2.93–8.53; n = 22) | p = 0.216 |

**Note:**
Units for peak amplitude are dimensionless sample values.

**Table 4 Comparison of call characteristics measured for *Eleutherodactylus jamesdixoni* and *E. nitidus*.** Means between species were compared using a Student's *t*-test.

| | E. jamesdixoni sp. nov. (N = 3) | E. nitidus (N = 4) | p value |
|---|---|---|---|
| Call duration (s) | 0.57 ± 0.06 (0.47–0.71; n = 45) | 0.27 ± 0.04 (0.21–0.35; n = 36) | p < 0.001 |
| Call rise time (s) | 0.08 ± 0.01 (0.04–0.36; n = 45) | 0.02 ± 0.00 (0.01–0.04; n = 36) | p < 0.001 |
| Peak amplitude | 19779 ± 3763 (10964–31962; n = 45) | 20902 ± 2634 (11479–30961; n = 36) | p = 0.659 |
| Intercall interval (s) | 23.09 ± 2.46 (17.71–48.78; n = 41) | 7.97 ± 2.04 (4.06–20.85; n = 32) | p < 0.001 |
| Dominant frequency (kHz) | 2.58 ± 0.017 (2.50–2.67; n = 45) | 2.48 ± 1.52 (2.24–2.67; n = 36) | p = 0.325 |
| Minimum fundamental frequency (kHz) | 2.50 ± 0.033 (2.41–2.67; n = 45) | 2.39 ± 0.13 (2.24–2.58; n = 36) | p = 0.204 |
| Maximum fundamental frequency (kHz) | 2.78 ± 0.03 (2.76–2.84; n = 45) | 2.70 ± 0.17 (2.50–2.84; n = 36) | p = 0.472 |
| Pulse duration (s) | 0.03 ± 0.00 (0.01–0.09; n = 639) | NA | NA |
| Pulse peak amplitude | 9982 ± 1035 (637–31962; n = 639) | NA | NA |
| Interpulse interval (s) | 0.05 ± 0.01 (0.01–0.07; n = 131) | NA | NA |
| Pulse dominant frequency (kHz) | 2.63 ± 0.032 (2.41–2.84; n = 639) | NA | NA |
| Pulse minimum fundamental frequency (kHz) | 2.54 ± 0.03 (0.09–2.76; n = 639) | NA | NA |
| Pulse maximum fundamental frequency (kHz) | 2.72 ± 0.03 (2.50–2.84; n = 639) | NA | NA |

**Note:**
Units for Peak amplitude and Pulse peak amplitude are dimensionless sample values.

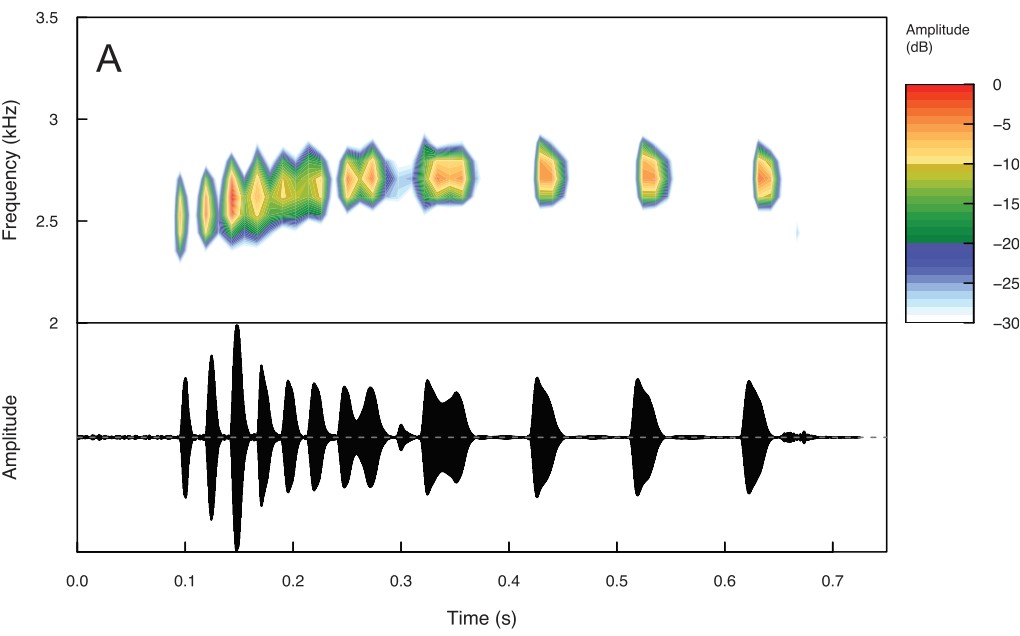

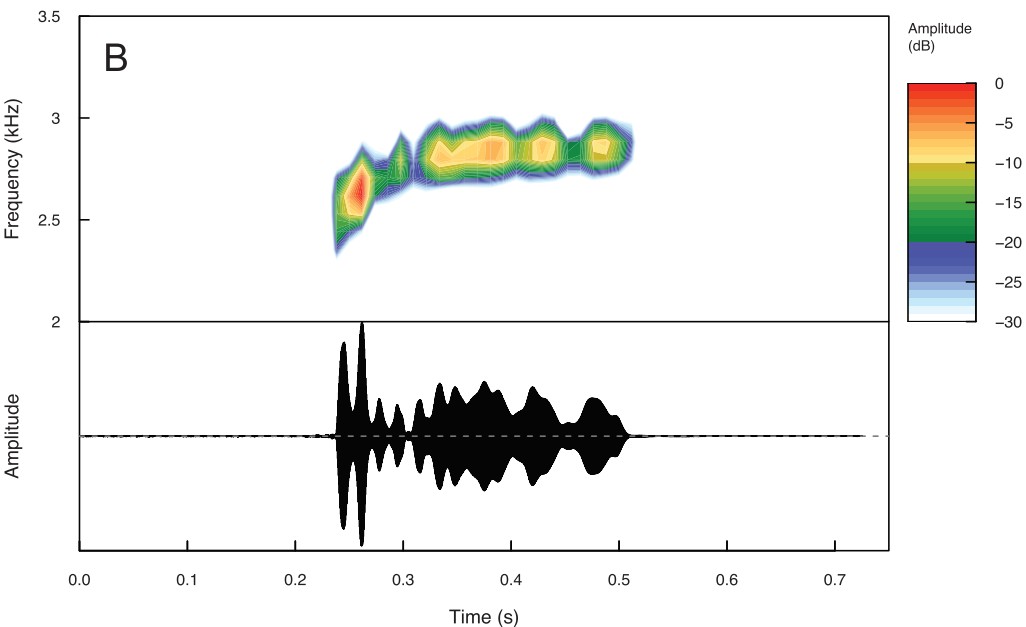

**Figure 6 Spectrograms and oscillograms of one call from (A) a paratype (IBH 34851) of** *Eleutherodactylus jamesdixoni* **in comparison to (B) a topotypic** *E. nitidus* **sample (IBH 34809) showing differences in temporal and spectral properties.**

Raw and filtered call recordings are available in the Dryad digital repository associated with this article (*Devitt et al., 2023*). Sample calls for each species are also available on AmphibiaWeb (https://amphibiaweb.org/).

## Species delimitation

Based on combined evidence from DNA sequences, morphology, and male advertisement calls, we name two new species below. We describe coloration and provide color photos in life and in preservative. We discuss morphological features that may be useful in distinguishing between close relatives and provide call recordings to facilitate identification. The new species—like many new species of terraranan frogs—were first discovered using DNA sequences. To facilitate future species delimitation in this clade, we use molecular synapomorphies for diagnosis in addition to descriptions of morphological features.

>*Eleutherodactylus* (*Syrrhophus*) *humboldti*, new species
>LSID urn:lsid:zoobank.org:act:8EB140BE-9DAE-46A3-853D-7FF9AB306CB0
>**Suggested English name:** Humboldt's Peeping Frog
>**Suggested Spanish name:** Rana fisgona de Humboldt

## Holotype (Fig. 7)

MZFZ 4505 (field no. TJD 1307), adult male from Mexico, State of México, Temascaltepec Municipality, Rancho El Pinal (19.09433, −100.08279), 2,428 m, collected by Thomas J. Devitt, Antonio Esaú Valdenegro Brito, Rodrigo Gabriel Martínez Fuentes, and Peter Heimes on June 27, 2022.

## Paratypes (*n* = 3; Fig. 7)

MZFZ 4503–4 (field nos. TJD 1305–6), both adult males from the same locality collected by the same individuals on the same date as the holotype; MZFZ 4506 (field no. TJD 1308), adult male from Mexico, State of México, Temascaltepec Municipality, Hotel Las Piñas Avándaro (19.15383, −100.14292), 1,982 m, collected by the same individuals on the same date as the holotype.

## Referred specimens (Fig. 7)

IBH 34816–8 (field nos. TJD 833–5), all adult males from Mexico, State of México, Municipality of Temascaltepec, Rancho El Pinal (19.09000, −100.07779), 2,245 m, collected by Uri Omar García Vázquez on June 28, 2011.

## Diagnosis and comparisons

Molecular synapomorphies for this species are (character: state change): 147: A => G, 204: C => T, 241: C => T, 243: A => T, 256: T => C, 279: C => A, 281: C => T, 292: G => A, 295: A => G and 358: G => C. Character numbers refer to the position in the alignment (Data S1).

Among species of the western clade of *Syrrhophus*, *E. humboldti* is most similar in appearance to *E. maurus* (Fig. 8); however, our sample sizes for each species are too small to make robust inter- or intrapopulation comparisons. These two taxa are very similar in having a triangular head with a sharp canthus rostralis with a black stripe on the lateral aspect. The stripe fades inferiorly into a grayish brown. This stripe seems to extend across the eye in the sense that the middle and lower part of the iris is black, whereas the upper

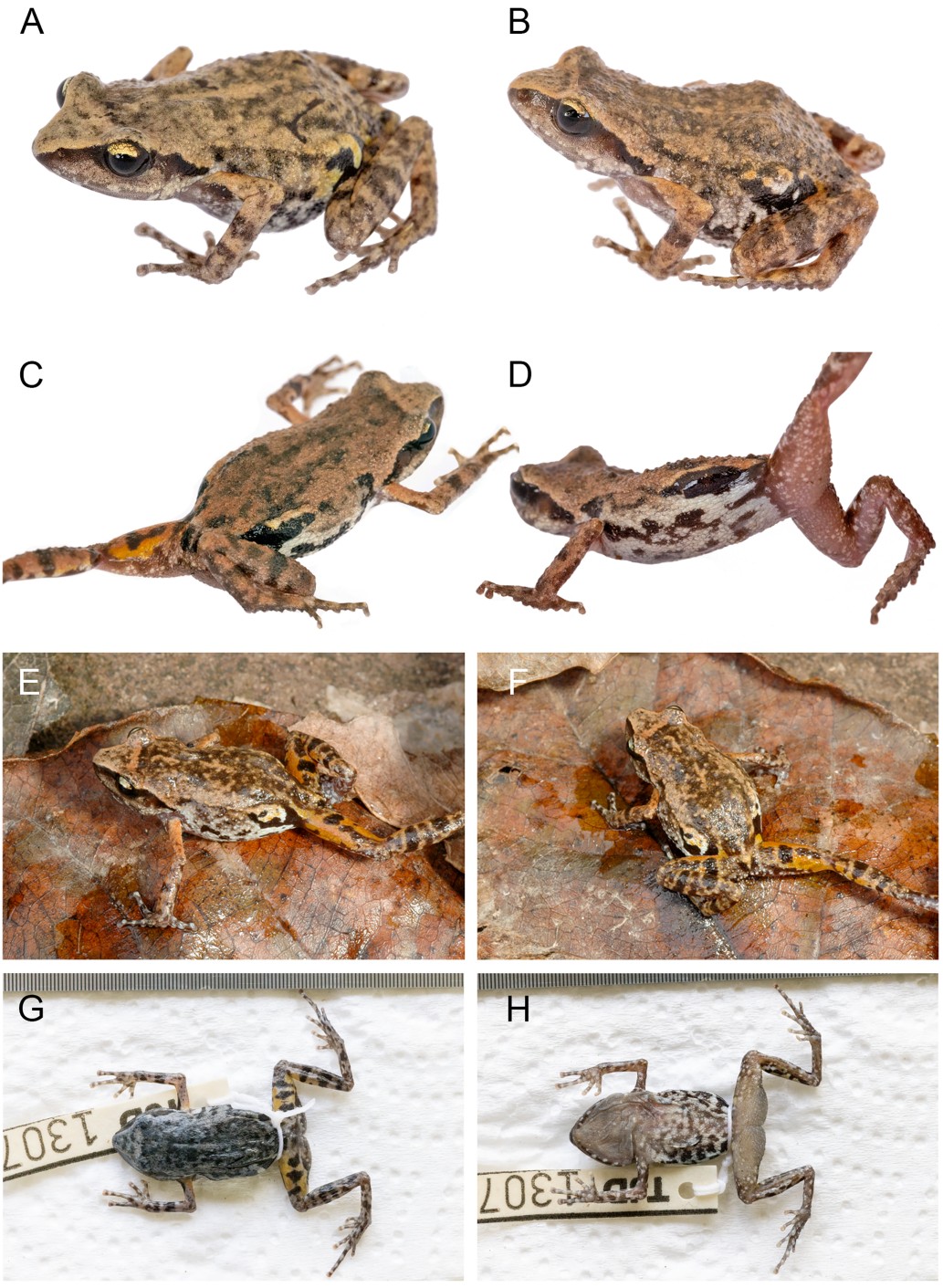

**Figure 7 *Eleutherodactylus humboldti*, sp. nov. in life and preservative.** (A) MZFZ 4505 (holotype); (B) MZFZ 4504 (paratype); (C and D) MZFZ 4506 (paratype); (E and F) IBH 34816; (G and H) MZFZ 4505 (holotype) in preservative.

part of the iris is gold. The stripe extends posterior to the eye across the tympanum (which is slightly paler in color) to the axilla. In the postorbital region, the stripe borders the sharp supratympanic fold.

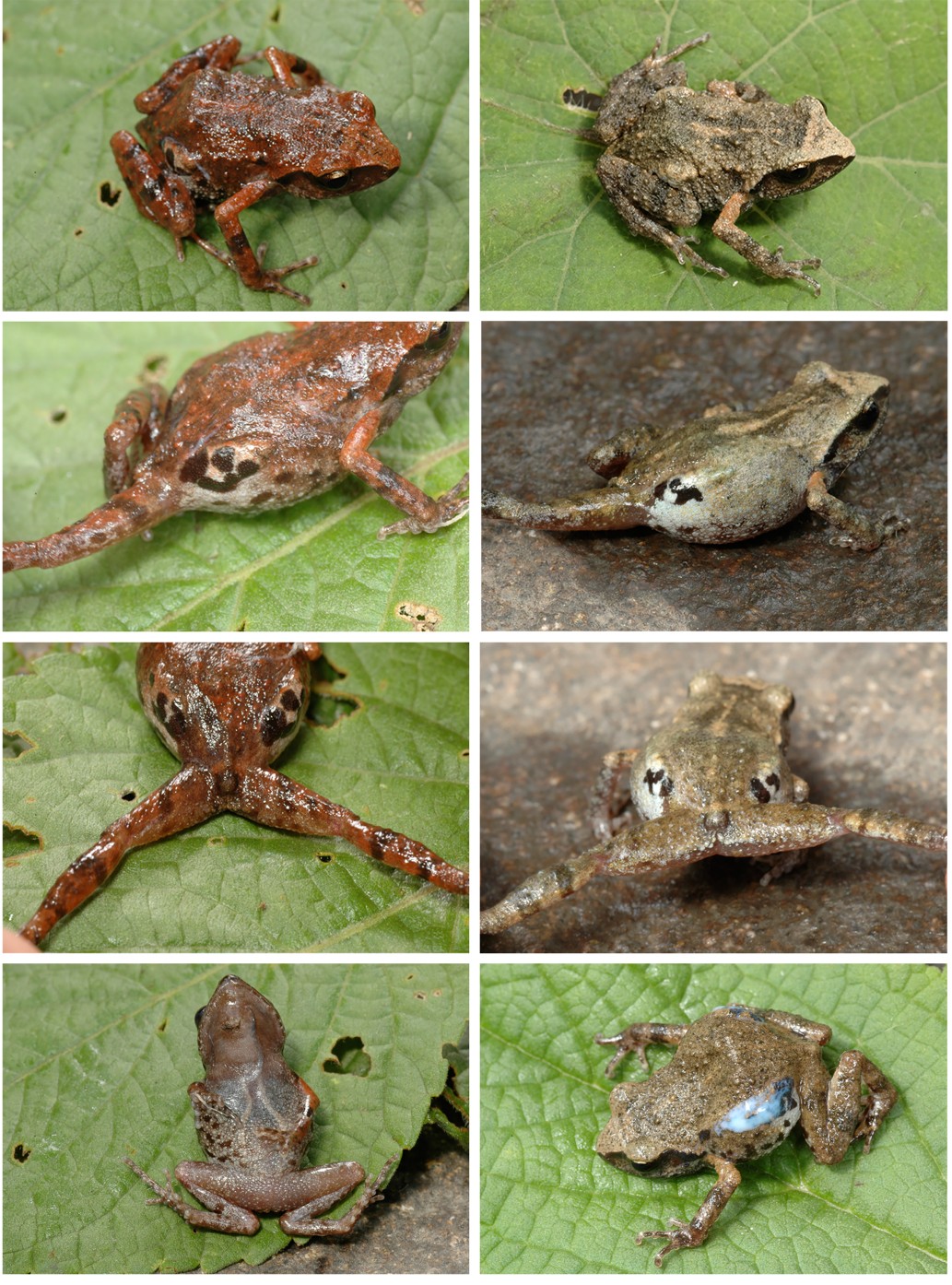

**Figure 8** *Eleutherodactylus maurus* from the type locality showing color polymorphism in two individuals (lefthand column: IBH 34838; righthand column: IBH 34839).

The dorsum of *E. humboldti* is brownish tan, and in *E. maurus* it is either brownish tan or rust-colored with a few scattered dark brown or black markings. Both color morphs of *E. maurus* occur in the same population at the type locality for that species (Fig. 8). The snout lacks markings, is distinctly paler than the dorsum, and its color is set off

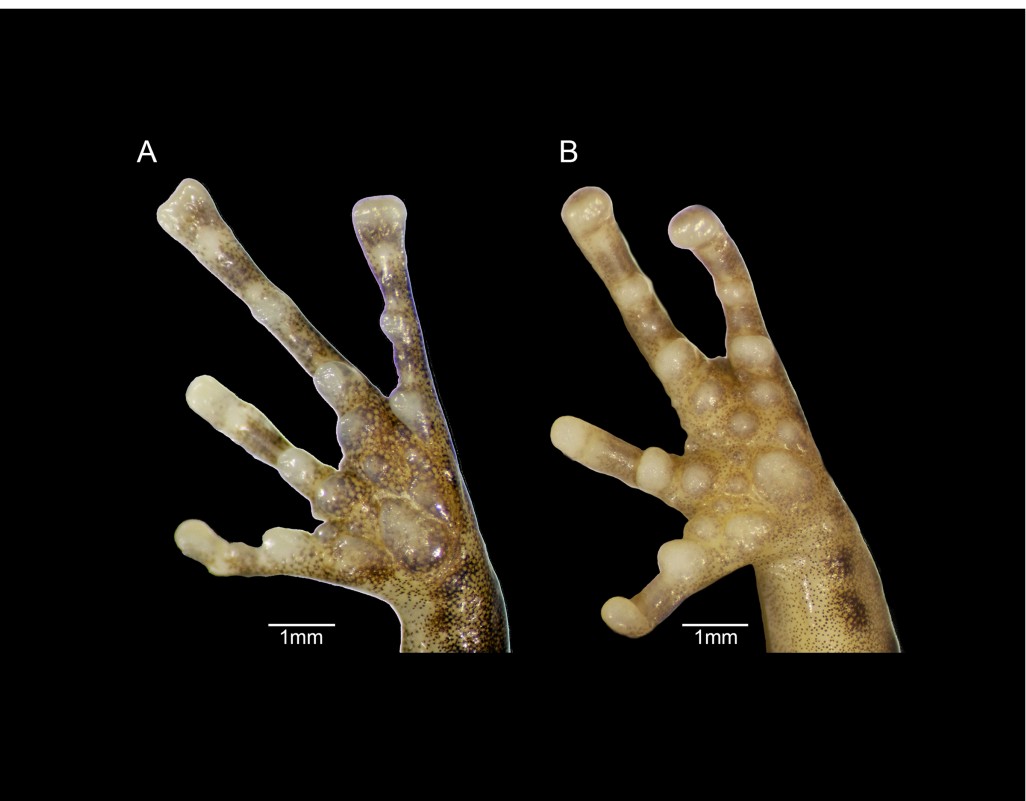

**Figure 9** Forefeet of (A) *Eleutherodactylus humboldti* sp. nov. (IBH 34818) and (B) *E. maurus* (IBH 34838).

abruptly from the dorsal color in the midorbital region; however, there is no distinct pale interorbital line as there is in *E. jamesdixoni* and *E. nitidus*.

Both species have scattered tubercles on the dorsal surfaces of the body and limbs, but those in *E. maurus* are slightly larger. In both a large gland is present in the inguinal region. The skin covering the gland is mottled black and white, with black predominating. The dorsal color extends onto the flanks, fading into a pale gray with irregular blackish blotches. The blotches are larger and more extensive in *E. humboldti* and the blackish markings are absent midventrally in *E. maurus*. The color of the dorsal surface of the forearm (but not the upper arm) and hindlimbs is like that of the dorsum. The color of the concealed surfaces differs between the two species: *E. humboldti* has an orange suffusion of the posterior thigh and upper arm (Fig. 7) that is lacking in *E. maurus*.

The ventral surfaces of the chin, belly, and limbs are translucent and tan to brownish. The posterior belly and thighs have uniform small tubercles, some of which have white tips.

The fingers of *E. humboldti* are slenderer and longer than those of *E. maurus* (Fig. 9). The tips of the third and fourth fingers of *E. humboldti* are slightly wider than those of *E. maurus* (*E. humboldti*: Fin3DW = 0.857 ± 0.099 mm, Fin4DW = 0.806 ± 0.066 mm; *E. maurus*: Fin3DW = 0.718 ± 0.126 mm; Fin4DW = 0.645 ± 0.095 mm; Fig. 9). The hand of *E. maurus* is slightly more robust than that of *E. humboldti*; the subarticular tubercles of

*E. maurus* are larger and more prominent, and the supernumerary tubercles are more numerous. The palmar tubercle of *E. maurus* is smaller and more rounded (Fig. 9).

Other species of *Eleutherodactylus* that occur in this general region of Mexico include *E. angustidigitorum* (*Taylor, 1940b*), *E. floresvillelai* (*Grünwald et al., 2018*), *E. grandis*, *E. nitidus*, and *E. petersi*. Of these species, only *E. petersi* possesses third and fourth fingers that have expanded tips. The tips of these digits are wider in *E. petersi* than in *E. humboldti* and *E. petersi* has a mottled dorsal color pattern and silvery white spotting on the upper lip (*Dixon, 1957*) that *E. humboldti* lacks.

### Measurements of the holotype (Fig. 7)

Adult male (MZFZ 4505). Measurements (in mm) are as follows: HW 9.599, SVL 24.638, TL 10.154, IOD 3.406, ED 2.621, IND 1.711, EN 1.814, FL 9.327, TD 0.926, THL 9.727, SL 2.833, HAL 5.63, FLL 6.552, Fin3L 6.114, Fin3DW 0.939, Fin4L 5.005, and Fin4DW 0.859. Vocal slits are present. Advertisement call recordings (raw and filtered) and associated output files are available in the Dryad digital repository for this article (*Devitt et al., 2023*).

### Coloration in life (Fig. 7)

The dorsum is brownish tan. The snout lacks markings and is paler than the dorsum, set off from the dorsal color in the midorbital region. The dorsal color extends onto the flanks, fading into a pale gray with irregular blackish blotches. The skin covering the inguinal gland is mottled black and white, with black predominating. The dorsal surface of the forearm (but not the upper arm) and hindlimbs is colored like that of the dorsum. The concealed surfaces of the posterior thigh and upper arm are suffused with orange. The ventral surfaces of the chin, belly, and limbs are translucent and tan to brownish. The posterior belly and thighs have uniform small tubercles, some of which have white tips.

### Coloration in preservative

In preservative, the dorsum fades to gray with dark gray blotches, and the orange suffusion of the concealed surfaces fades to pale yellowish gray. The belly fades to light gray with dark gray blotches and the chin fades to grayish lavender. Dorsal tubercles may not be visible in preservative.

### Variation

The type series plus three referred individuals are uniform in coloration (Fig. 7) and size (Data S3).

### Distribution and natural history

This species is known from three localities in the Quaternary Valle de Bravo volcanic field in the state of México in Central Mexico. This volcanic field is part of the Mil Cumbres physiographic subprovince within the Eje Neovolcánico physiographic province. The type series was collected at night between 21:00 and 0:00 during a thunderstorm with ambient air temperatures of 9–11 °C. Habitat at the type locality was a disturbed clearing in

pine-oak forest. Individuals were calling from low (<1.5 m) vegetation, frequently within dense bunches of non-native grass.

### Etymology

We name this species in honor of German scientist Alexander von Humboldt, whose year-long exploration of south-central Mexico at the turn of the 19th century resulted in the first scientific account of the New World and whose writings continue to influence biogeographers, ecologists and evolutionary biologists seeking to understand the origin and maintenance of biodiversity in mountain regions (*Rahbek et al., 2019b*, *2019a*).

> ***Eleutherodactylus* (*Syrrhophus*) *jamesdixoni*, new species**
> urn:lsid:zoobank.org:act:6DB31FC7-7DC2-413B-9934-1D370CF5D5EB
> **Suggested English name: Dixon's Peeping Frog**
> **Suggested Spanish name: Rana fisgona de Dixon**

### Holotype (Fig. 10)

IBH 34852 (field no. TJD 897), adult male from Mexico, State of Sinaloa, Municipality of Concordia, 7.7 road miles west of El Palmito on the highway from Concordia to Durango (23.50418, −105.83701), 1,992 m, collected by Thomas J. Devitt, Andrea Roth Monzón, and Andrés Alberto Mendoza Hernández on July 26, 2011.

### Paratypes

IBH 34849, 34851 (field nos. TJD 894 and TJD 896), both adult males from the same locality collected by the same individuals on the same date as the holotype.

### Referred specimens

UTA A-5905 (JAC 23704), UTA A-59506 (JAC 23782), TNHC 117001 (JSR 580A), and MZFC 4789 (OFV 325).

### Diagnosis and comparisons

Molecular synapomorphies for this species are (character: state change) 204: C => T, 291: C => A, and 360: C => A. Character numbers refer to the position in the alignment (Data S1).

Among species of the western clade of *Syrrhophus*, *E. jamesdixoni* (Fig. 10) is most similar in appearance to *E. nitidus* (Fig. 11). We restrict our comparison to topotypic samples of *E. nitidus*. The two species are very similar in having a roughly triangular head with a gently angled canthus rostralis. The lateral aspect of the canthus is darker than the dorsal surface of the snout. In *E. jamesdixoni* a dark greenish black stripe is present, but in *E. nitidus* this stripe varies from very dark to barely differentiated from the color of the snout (Fig. 11). The stripe fades inferiorly in *E. nitidus*, but in *E. jamesdixoni* it is more distinct. This stripe seems to extend across the eye in that the middle and lower part of the iris is dark, whereas the upper part of the iris appears as densely packed gold flecks (this golden region is larger than in *E. maurus* and *E. humboldti*). In both, the stripe extends posterior to the eye across the tympanum (which is slightly copper-colored) to the axilla

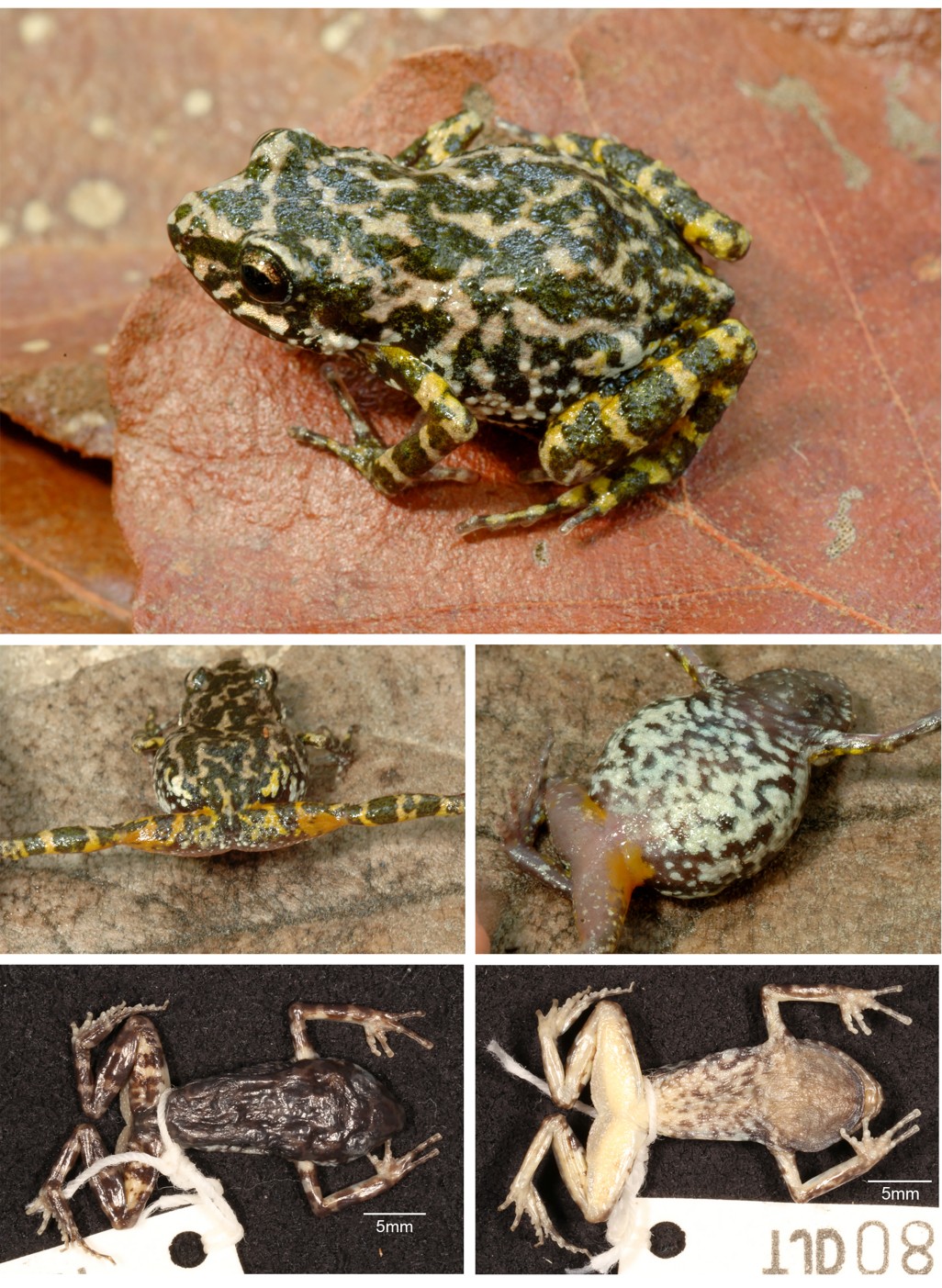

**Figure 10 Holotype of *Eleutherodactylus jamesdixoni*, sp. nov. (IBH 34852) in life and preservative.**

but is much less distinct than in *E. maurus* and *E. humboldti*. In the postorbital region the stripe becomes indistinct. The supratympanic fold is rounded but weakly developed.

In *E. jamesdixoni* the dorsal surfaces of the head, body, and limbs have a contrasting mottled pattern of dark green and tan to pale yellow-orange markings. The dorsal surfaces of *E. nitidus* are more variable; some have contrasting markings of tan and dark brown

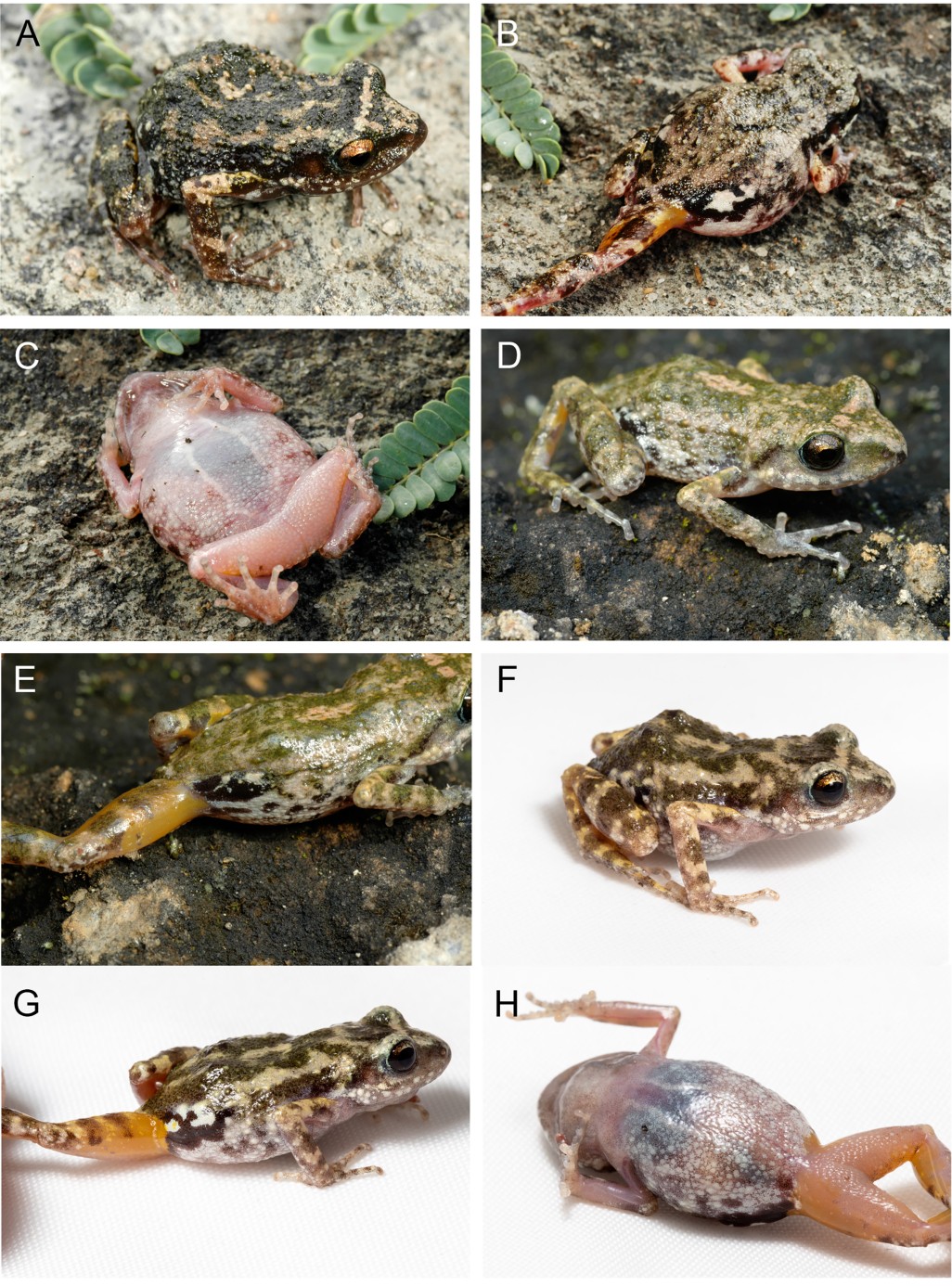

**Figure 11** *Eleutherodactylus nitidus* **in life.** (A) IBH 34810 and (B and C) IBH 34812 from the type locality (Izucar de Matamoros, Puebla); (D and E) IBH 34788 from Guerrero; (F–H) MZFZ 4486 from Oaxaca. See Table S1 for specific localities.               

(Fig. 11), but others have tan to dark brown or pale greenish dorsal surfaces with few contrasting markings. In both species (and in contrast to *E. maurus* and *E. humboldti*), the

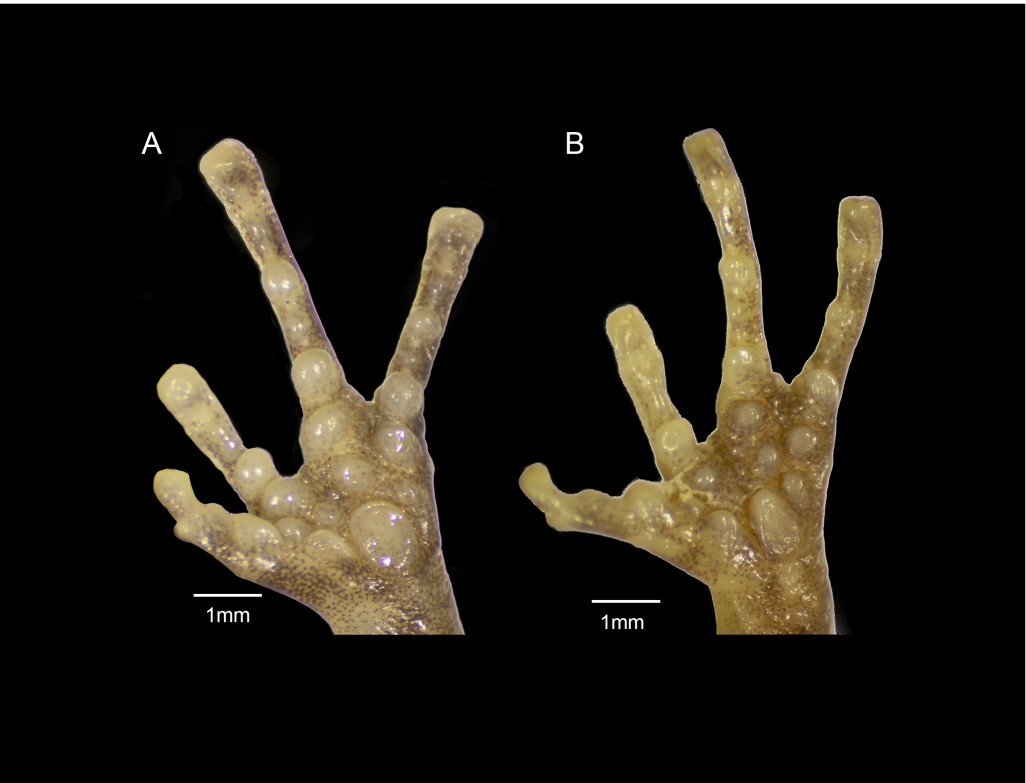

**Figure 12 Forefeet of (A)** *Eleutherodactylus jamesdixoni* **sp. nov. (IBH 34851) and (B)** *E. nitidus* **(IBH 34812).**

dorsal surfaces of the snout are patterned like the dorsum. The snout and dorsum are separated by a distinct pale interorbital line.

Both species have tubercles irregularly scattered on the dorsal surfaces of the body and limbs, but those in *E. nitidus* are slightly larger. In both, a gland is present in the inguinal region; it is smaller and less protuberant than in *E. maurus* and *E. humboldti*. The skin covering the gland is mottled black and white, with black predominating. Because of the strong contrast in the pattern, the gland is less obvious in *E. jamesdixoni*.

In both species the pattern and color of the dorsum extends onto the flanks. In *E. nitidus* it fades to a pale gray with irregular blackish markings. In *E. jamesdixoni* the flanks and belly are mottled with dark gray and white, whereas in *E. nitidus* the belly is pale, translucent, and lacks markings.

The color of the concealed surfaces is similar in both species. An orange suffusion is present on the posterior thigh, anterior thigh and adjacent inguinal region, and upper arm (Figs. 10 and 11).

In *E. nitidus* the ventral surfaces of the chin, belly, and limbs are translucent and vary from grayish pink, apparently lacking pigment, to dull gray. In both species, the posterior belly and thighs have uniform small tubercles, some of which have white tips.
In *E. jamesdixoni* the belly and portions of the ventral sides of the limbs are mottled with black and white and the chin is darkly pigmented.

The general shape and relative sizes of the fingers, including the expansion of digital tips, are similar in both species (Fig. 12). The subarticular, palmar, and supernumerary tubercles of *E. nitidus* are slightly more pronounced than those of *E. jamesdixoni*. The palmar tubercle of *E. nitidus* is more elongate and conical than that of *E. jamesdixoni* (Fig. 12).

Other species of *Eleutherodactylus* that occur within the range of this species include *E. interorbitalis* (*Langebartel & Shannon, 1956*), *E. modestus*, *E. pallidus* (*Duellman, 1958*), *E. saxatilis* (*Webb, 1962*), *E. teretistes* and *E. wixarika* (*Reyes-Velasco et al., 2015*). Of these species, only *E. saxatilis* possesses a compact lumbar or inguinal gland. *Eleutherodactylus saxatilis* is easily distinguished from *Eleutherodactylus jamesdixoni* by third and fourth fingers that are truncate, with tips about twice the width of the narrowest part of the digit.

## Measurements of holotype

Adult male (IBH 34852). Measurements (in mm) are as follows: HW 8.785, SVL 21.961, TL 8.709, IOD 2.585, ED 2.280, IND 0.913, EN 1.667, FL 8.672, TD 0.969, THL 8.635, SL 2.817, HAL 5.254, FLL 6.655, Fin3L 5.432, Fin3DW 0.698, Fin4L 4.38, and Fin4DW 0.563. Vocal slits are present. Advertisement call recordings (raw and filtered) and associated output files are available in the Dryad digital repository for this article (*Devitt et al., 2023*).

## Coloration in life (Fig. 10)

The dorsal surfaces of the head, body, and limbs have a contrasting mottled pattern of dark green and tan to pale yellow-orange markings. A distinct pale interorbital line separates the snout from the dorsum. The belly and portions of the ventral sides of the limbs are mottled with black and white; the chin is darkly pigmented. The skin covering the inguinal gland is mottled black and white, with black predominating. The pattern and color of the dorsum extends onto the flanks. The concealed surfaces of the posterior thigh, anterior thigh and adjacent inguinal region, and upper arm are suffused with orange (Figs. 10 and 11). A greenish black stripe is present along the lateral aspect of the canthus. The lower portion of the iris is dark, whereas the upper part of the iris is golden. The stripe extends posterior to the eye across the copper-colored tympanum to the axilla but becomes indistinct in the postorbital region.

### Coloration in preservative (Fig. 10)

In preservative, the dorsum fades to dark gray, and the orange suffusion of the concealed surfaces fades to a light cream color. The belly fades to a cream background color with dark grayish lavender blotches and the chin fades to grayish lavender. Dorsal tubercles may not be visible in preservative.

## Variation

The type series is uniform in coloration and size (Data S3).

## Distribution and natural history

*Eleutherodactylus jamesdixoni* is found in the southern Sierra Madre Occidental and westernmost portions of the Eje Neovolcánico and Sierra Madre del Sur physiographic provinces. The type series was collected at night in pine-oak woodland between 21:30 and

23:50 at 15°C. Individuals were calling from low vegetation perched approximately 0.5 m above ground. This species is sympatric with *E. saxatilis* at the type locality, a saxicolous species found on rock outcrops.

### Etymology

We name this species in honor of the late James R. Dixon, Professor Emeritus and Curator Emeritus of amphibians and reptiles at the Biodiversity Research and Teaching Collections at Texas A&M University. Dr. Dixon provided the first and most thorough statistical analysis to date of geographic variation among populations of the former genus *Tomodactylus* from western Mexico (*Dixon, 1957*).

## DISCUSSION

Here, we describe two new species of *Eleutherodactylus* (*Syrrhophus*) from western and central Mexico based on genetic, morphological, and advertisement call differences. *Eleutherodactylus humboldti* of the Quaternary Valle de Bravo volcanic field is about 3% divergent in 16S from its sister taxon *E. maurus*. *Eleutherodactylus jamesdixoni* of the Sierra Madre Occidental and westernmost portions of the Sierra Madre del Sur and Eje Neovolcánico is also about 3% divergent from its sister taxon *E. nitidus*. Differences in color pattern, the size and shape of the fingers, and advertisement calls allow these species to be distinguished.

Our work expands on previous efforts to infer evolutionary relationships within *Syrrhophus* and make taxonomic changes that more accurately reflect species level diversity (*Grünwald et al., 2018*, *2021*; *Hernández-Austria et al., 2022*). Most species were recovered as well-supported monophyletic groups in our phylogeny, with two exceptions. First, *E.* "*angustidigitorum*" (comprising three lineages numbered with suffixes 1–3) is paraphyletic with respect to *E. grandis* and *E. floresvillelai*. We sampled *E. grandis* from the type locality and included published sequences of topotypic *E. floresvillelai*, but because we did not sample from the type locality of *E. angustidigitorum* it remains unclear which of the three lineages (if any) represents true *E. angustidigitorum*. Published sequences assigned to *E. angustidigitorum* (*Grünwald et al., 2018*, *2021*; *Hernández-Austria et al., 2022*) are also not from the type locality for this species and should be reexamined.

Second, the group of *E. pipilans* sequences is paraphyletic to a clade containing a mixture of *E. rubrimaculatus* and *E. nebulosus* sequences. We sampled from the type locality of *E. pipilans* ("14.6 km. S Mazatlán, Guerrero, México"), and included published sequences from topotypic samples of *E. rubrimaculatus* and specimens identified as *E. nebulosus* (*Hernández-Austria et al., 2022*). These species occur in the same general region (*Taylor & Smith, 1945*) and may come into contact, but their boundaries remain unclear. *Grünwald et al. (2021)* concluded that *E. rubrimaculatus* was a junior synonym of *E. nebulosus* because they found the former taxon to be paraphyletic with respect to *E. nebulosus*, and the two exhibited low genetic divergence. *Hernández-Austria et al. (2022)*, however, questioned this conclusion because *Grünwald et al. (2021)* had not sampled topotypic specimens, and recognized these taxa as distinct despite the low divergence. Further work using nuclear sequence data, morphology, and advertisement

calls is needed to determine whether these taxa are distinct species. We urge future workers to avoid making taxonomic changes based on mitochondrial DNA alone and without examining type specimens.

## CONCLUSIONS

Here, we have delimited two new species of *Syrrhophus* using morphology, calls, and DNA-based diagnoses. These frogs are among the most ubiquitous and abundant of Mexican amphibians, yet remain among the least known. Exceptional microendemism typifies this group, with several species known only from the type locality. Their ubiquity, diversity, high endemism, and unusual reproductive mode (direct development) offer potential for addressing fundamental and emerging questions in ecology, evolution, and behavior. This potential has not yet been realized however, due to an inaccurate taxonomy that underestimates species-level diversity. Additional species await description; the need for this research is great, and the timing is urgent. Mexico is home to two Biodiversity Hotspots—the Madrean Pine Oak Woodlands and tropical lowlands of Mesoamerica—but high rates of deforestation and a general lack of conservation areas pose increasing threats to species survival there. Some 60% of Mexican amphibians are threatened with or have already been lost to extinction, including nine species of *Syrrhophus* that are classified as Endangered or Critically Endangered by the International Union for the Conservation of Nature (*IUCN, 2021*). Discovering and describing unrecognized species-level diversity and identifying areas of endemism will directly impact conservation prioritization in Mexico.

## ACKNOWLEDGEMENTS

We thank Adrián Nieto Montes de Oca and Oscar Flores Villela for donations of tissues and allowing us to examine specimens in their care in the Herpetology Collection of the Faculty of Sciences "Alfonso L. Herrera" Museum of Zoology (MZFC) at the National Autonomous University of Mexico (UNAM). For assistance with specimen loans and cataloging at the MZFC, we thank Edmundo Pérez-Ramos. We thank Gabriela Parra Olea and Omar Hernández Ordóñez for allowing us to examine specimens in their care in the National Collection of Amphibians and Reptiles at the Biology Institute (CNAR IBH) at UNAM. We thank Uri Omar García-Vázquez, Jose Jesús Sigala-Rodriguez, Raquel Hernández Austria, and Sean M. Rovito for donation of tissues. We thank Jonathan A. Campbell, Matthew K. Fujita, Eric N. Smith, and Greg Pandelis for donations of tissues and allowing us to examine specimens in their care at the University of Texas at Arlington. We owe special thanks to Jonathan A. Campbell, Eric N. Smith, and Oscar Flores Villela for inviting the first author on collecting trips beginning in 2003 as part of their biodiversity survey and inventory work in Mexico. For specimen loans and photographs of type specimens, we thank Anita Bendict (Mayborn Museum, Baylor University), Greg Schneider (UMMZ), Alan Resetar and Joshua Mata (FMNH), Chris Phillips (UIMNH), José Rosado and Joseph Martinez (MCZ), Carol Spencer and Jim McGuire (MVZ), Kelsey Minatra and Travis J. LaDuc (TNHC). For assistance in the field, we thank Uri Omar García-Vázquez, Antonio Esaú Valdenegro Brito, Rodrigo Gabriel Martínez Fuentes, Andrea Roth Monzón, Andrés Alberto Mendoza Hernández Eric Centenero Alcala, Itzue

Caviedes Solis, Peter Heimes, Sean M. Rovito, John D. McVay, Jorge Fernando Sánchez Solís, Leticia Ochoa Ochoa, and Carlos Alberto Hernández Jiménes.

### Funding
This work was supported by a UC MEXUS-CONACyT Postdoctoral Fellowship to Thomas J. Devitt and a Mohamed bin Zayed Species Conservation Fund Award (Project Number 13252103) to Thomas J. Devitt. The funders had no role in study design, data collection and analysis, decision to publish, or preparation of the manuscript.

### Grant Disclosures
The following grant information was disclosed by the authors:
UC MEXUS-CONACyT Postdoctoral Fellowship.
Mohamed bin Zayed Species Conservation Fund Award: 13252103.

### Competing Interests
The authors declare that they have no competing interests.

### Author Contributions
- Thomas J. Devitt conceived and designed the experiments, performed the experiments, analyzed the data, prepared figures and/or tables, authored or reviewed drafts of the article, and approved the final draft.
- Karen Tseng performed the experiments, analyzed the data, prepared figures and/or tables, authored or reviewed drafts of the article, and approved the final draft.
- Marlena Taylor-Adair performed the experiments, analyzed the data, prepared figures and/or tables, authored or reviewed drafts of the article, and approved the final draft.
- Sannidhi Koganti performed the experiments, analyzed the data, prepared figures and/or tables, and approved the final draft.
- Alice Timugura performed the experiments, analyzed the data, prepared figures and/or tables, and approved the final draft.
- David C. Cannatella conceived and designed the experiments, analyzed the data, prepared figures and/or tables, authored or reviewed drafts of the article, and approved the final draft.

### Animal Ethics
The following information was supplied relating to ethical approvals (*i.e.*, approving body and any reference numbers):
The Institutional Animal Care and Use Committee at the University of Texas at Austin approved this research (AUP-2022-00192).

### Field Study Permissions
The following information was supplied relating to field study approvals (*i.e.*, approving body and any reference numbers):

Collecting permits in Mexico were issued by the Secretaría de Medio Ambiente y Recursos Naturales (SEMARNAT).

## Data Availability

The sequences are available from GenBank (OQ145174–OQ145320, OQ311392–OQ311403, OQ408142–OQ408148) and in Supplemental data S1.

The raw measurements and R scripts are available in the Supplemental Files.

The acoustic recordings, scripts, and photos of preserved specimens that were measured are available at the Dryad digital repository associated with this article (Devitt et al., 2023) at https://doi.org/10.5061/dryad.kprr4xh8f.

## New Species Registration

The following information was supplied regarding the registration of a newly described species:

Publication LSID: urn:lsid:zoobank.org:pub:C59B3FB8-839F-407C-ADC0-67C1FEA6A5BB

Eleutherodactylus LSID: urn:lsid:zoobank.org:act:734FAF15-32EF-42FB-893E-9E8AC91B28DD

*Eleutherodactylus humboldti* LSID: urn:lsid:zoobank.org:act:8EB140BE-9DAE-46A3-853D-7FF9AB306CB0

*Eleutherodactylus jamesdixoni* LSID: urn:lsid:zoobank.org:act:6DB31FC7-7DC2-413B-9934-1D370CF5D5EB.

## Supplemental Information

Supplemental information for this article can be found online at http://dx.doi.org/10.7717/peerj.14985#supplemental-information.

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
