# Peer review of "Two new species of Eleutherodactylus from western and central Mexico (Eleutherodactylus jamesdixoni sp. nov., Eleutherodactylus humboldti sp. nov.)"

_PeerJ, doi:10.7717/peerj.14985_

## Round 0.1 · original submission · Minor Revisions

I have read the paper written by Devitt et al on the description of two new species of frigs from Mexico. I agree with both reviewers that a minor revision is required before the paper is ready to be accepted. I especially agree with rev 1 in that the diagnosis should be expanded to include geographically close species because the likelihood of the new species occurring in new populations is high.

Please pay close attention to all the comments made by the reviewers, as I find all of them valid. Taxonomically, this is a very difficult group of frogs and the community will greatly benefit from the improved version of this manuscript.

Reviewer 1 ·

Basic reporting

This paper is well written and sensibly structured, with sufficient and relevant citations. Both the introduction and the conclusion provide important taxonomic context to the descriptions. The figures are in general well done although I have some specific comments (see below).

Experimental design

This manuscript presents original primary research and contains a lot of new DNA sequence and morphological data. It addresses an important taxonomic problem in this understudied group and the authors took the vital step of examining and including type or topotypic material in the analyses. The taxonomy is done in an integrative way, including color data, morphology and calls, and all raw data are given. Enough detail is provided (along with an R script) that these analyses could be repeated for future species descriptions.

Validity of the findings

Taken as a whole, I found the descriptions provide convincing evidence that both species are distinct. All the data are provided in repositories or in the supplementary files.

Additional comments

This paper uses a combination of morphology, color data, and call data to diagnose and describe two new species of direct-developing frogs from Mexico. The sampling for the group is thorough and an effort was made to examine and include type/topotypic material both in the morphology and the phylogeny, which is a strength of the paper. The manuscript is well-written and the figures are well done and informative (but see suggestions below). Taken as a whole, I found the descriptions provide convincing evidence that both species are distinct. Beyond a number of specific suggestions (given below by line number), I have a few more general suggestions. For both species, I think that the diagnosis should be expanded to at least geographically proximate species rather than only to each new species’ sister taxon. These diagnoses need not be as thorough but it would be helpful in case new localities for the species are found and they need to be distinguished from other species that occur in the area. I think that the call data are some of the most convincing evidence for the distinctiveness of each sister species pair, but not much emphasis is given to these data in the descriptions themselves. Would it be possible to incorporate these data in the diagnosis section for each species? Finally, I appreciate that the authors are trying to get away from some of the traditional characters used in taxonomy of this group (ratios of disc width to finger width, etc). I think, however, that some of the morphological differences that are given could be quantified, even if the ranges are somewhat overlapping between sister species. In taxonomically difficult groups (particularly of small animals), it may be the case that only a suite of characters together can diagnose a species, but it seems here that there are some real differences in digit shape/width and tubercule number/size/shape that could be quantified more accurately.

This may be a matter of personal preference, but I’m not sure that much is gained by listing the specific molecular synapomorphies for each species, especially because the gene used was 16S. This gene obviously has no codons and commonly has indels, so the alignment can vary with each additional species included (or by choice of outgroup). This means that these synapomorphies will be hard to evaluate and compare in future phylogenies. I don’t have a strong objection to including them as a supplementary table, but I think that it would be sufficient to give a table of genetic distances between species (already included in supplementary material) without this level of detail about substitutions at particular sites.

Line 54: Given that Crawford and Smith (2005) started the process of dividing up Eleutherodactylus based on phylogenetic data, I think it would be appropriate to cite them here.

Lines 64–65: Were these two genera distinguishable morphologically, or only by geography?

Line 85: Spell out numbers less than ten (i.e. “eight”)

Line 104: Give permit number and state to whom permit was issued

Line 108: Is this the length of the alignment or the average length of the sequences? I ask because it is much shorter in salamanders using these primers

Line 135: Why were uncorrected p-distances calculated rather than using some model to obtain a more accurate distance estimate? Was this to compare with previously published distances for the group? If not, I would suggest using some sort of distance (i.e. GTR or even K2P) that corrects for multiple substitutions at a site

Line 182: It would be good to give the definitions here so that the reader doesn’t have to refer to the Köhler (2017) paper to know what they are.

Line 122: It is impossible to see the details of the phylogeny in the figure included in the PDF; the text is to small and blurry to read. I suggest splitting this figure up into two vertically so that the phylogeny is legible in the figure.

Line 310 and elsewhere: This may be just a personal preference, but I always write the name of the state as “México”. I agree that “Mexico” shouldn’t have an accent when the paper is in English but the names of Mexican states don’t have English versions so I write them as they are written in Mexico. The authors can choose to follow this or not, but I also think it helps to distinguish between the state and country.

Line 351: I suggest quantifying this difference using the measurements that were taken. I understand that both sample sizes are quite small but if the difference is real, it should show up in the measurements. This would help someone trying to identify frogs in the lab or field who isn’t able to sequence DNA.

Line 365: What about all of the other parts of the frog (venter, flanks, concealed surfaces)?

Line 402: Here and for the previous species, I found it hard to compare the color between the two species because they are in separate figure. Would it be possible to rearrange the figures so that there are side-by-side comparisons of each sister species pair showing the most important color differences?
Line 405: Varies how?

Lines 436–438: How do the tubercules look in jamesdixoni? Are they also present and, if so, are they similarly sized and colored?

Line 438: This difference in chin color seems like a diagnostic difference that would be useful for field identification. I would suggest limiting this section to differences between species and giving the full color description Coloration in life and Variation sections below. Otherwise the differences get somewhat lost.

Line 452: It is standard to give a textual color description even if this is shown in the figure. I think that this needs to be added here.

Line 487: line 471 specifically says that the phylogeny doesn’t include sequences from topotypic samples of rubrimaculatus and nebulosus, but here you say that you included Genbank sequences for topotypic material of these species. These statements are in direct conflict.

Lines 494–499: It isn’t clear to me exactly what this paragraph is trying to say. Giving the genetic divergence between nonsister pairs doesn’t seem that informative to me. The paragraph talks about intra- vs interspecific divergences but gives only values for interspecific divergences, and then doesn’t make much of a conclusion about anything. It would be valuable to add divergences between populations of what can clearly be said to be populations of the same species based on morphology and call data (if available). Failing that, it would be nice to at least give some intraspecific divergence values even if it’s not super clear if they are truly part of the same species. It would also be nice to put these values in more context. There are a lot of sequence data available for direct developing frogs (Craugastor, Pristimantis); how do the values on the lower end of divergence between known sister species pairs of Syrrhophus compare to those in other groups with a fairly similar natural history?

Line 500¬¬–503: This section is really brief and could use some additional information. Where is the type locality of petersi, and how far away are the specimens that were used in the phylogeny? Where are the other populations that are labeled as aff petersi? This is kind of an odd way to end the discussion, because it’s not a species that is discussed much in the rest of the paper and this paragraph is a bit on its own. Is there something more general that could be used to end the discussion? Maybe something about biogeography of the group or about the value of some kind of data used in this study (for example, calls) for taxonomic work in this group?

Line 505: The conclusion is usually a summation of the work, and I think a little bit more needs to be said related to the new species, if only because the paper is very focused on the two descriptions. The rest the conclusion is good and seems relevant to me.

Line 522: The inclusion of Guatemala here is odd because the rest of the paper deals only with Mexico. It’s fine to leave it here (this statement is true for essentially all tropical regions of the world, including Guatemala) but I would just say Mexico.

Figure 1: There are many colors in this figure that are essentially indistinguishable on the map. For similar colors, I think that a different shape should be used for each species so that they can be distinguished. For example, the colors for rufescens and humboldti are really similar and it’s not clear what points correspond to what speices without knowing a lot about Mexican geography and reading the text. Making one of the species triangles or squares would solve this.

Reviewer 2 ·

Basic reporting

This paper describes two new species of Eleutherodactylus frogs with a focus on diagnoses based on phylogenetic, morphological, and acoustic data. I thought the paper had sufficient background and methodological detail to understand the results and the conclusion. The manuscript is well-written and the raw data will be available via Dryad.

Experimental design

The paper is clear in its objective to describe two new species of Eleutherodactylus and the methods are consistent with the history of species delimitation and description in this genus of frogs. The methods used (phylogenetic, morphological analyses, and acoustic/frog call descriptions) are well-explained.

Validity of the findings

I thought that the data and results supported the objectives (descriptions of two new species of frogs) and I do not have comments for improvement.

Additional comments

I have just a few minor comments and clarifications.
Line 88: The authors make an interesting statement about not including topotypic specimens or the examination of type specimens and I would like to have seen a small discussion (a few sentences at most) of this as it relates to their study in the discussion.
Line 63: It is mentioned that up to four species of Syrrophus can be found in sympatry. Is it known if they hybridize? This is important given the authors use an evolutionary species concept. If it is not known, I would state that.
Line 107: I am assuming the "fluorometry" is the Qubit? Which version/kit?
Line 325: The authors state that they use DNA diagnoses as part of their description but I didn't see this in the descriptions.
Line 470: This is the first time the authors refer to their sampling for the study, and I might add a sentence about this in the methods. Also, how might the missing taxa affect their species delimitation?

---

## Round 0.2 · accepted · Accept

I have read the revised version of this manuscript and believe the authors did a great job paying attention to all of the reviewer's comments and modified the text accordingly where they thought appropriate.

I only have one more minor request. Please substitute field numbers for collection numbers in Fig 2 and give the meaning of the acronyms (both field numbers and museum or collection numbers) in the figure legend.